# Fast acting allosteric phosphofructokinase inhibitors block trypanosome glycolysis and cure acute African trypanosomiasis in mice

Iain W. McNae [1,8], James Kinkead[1,8], Divya Malik [1,8], Li-Hsuan Yen[1], Martin K. Walker[2], Chris Swain[3], Scott P. Webster[4], Nick Gray[1], Peter M. Fernandes [1], Elmarie Myburgh [5], Elizabeth A. Blackburn [1], Ryan Ritchie[6], Carol Austin[2], Martin A. Wear[1], Adrian J. Highton[2], Andrew J. Keats[2], Antonio Vong[2], Jacqueline Dornan[1], Jeremy C. Mottram [7], Paul A. M. Michels [1], Simon Pettit[2✉] & Malcolm D. Walkinshaw [1✉]

The parasitic protist *Trypanosoma brucei* is the causative agent of Human African Trypanosomiasis, also known as sleeping sickness. The parasite enters the blood via the bite of the tsetse fly where it is wholly reliant on glycolysis for the production of ATP. Glycolytic enzymes have been regarded as challenging drug targets because of their highly conserved active sites and phosphorylated substrates. We describe the development of novel small molecule allosteric inhibitors of trypanosome phosphofructokinase (PFK) that block the glycolytic pathway resulting in very fast parasite kill times with no inhibition of human PFKs. The compounds cross the blood brain barrier and single day oral dosing cures parasitaemia in a stage 1 animal model of human African trypanosomiasis. This study demonstrates that it is possible to target glycolysis and additionally shows how differences in allosteric mechanisms may allow the development of species-specific inhibitors to tackle a range of proliferative or infectious diseases.

[1] Wellcome Centre for Cell Biology, School of Biological Sciences, University of Edinburgh, Michael Swann Building, Max Born Crescent, Edinburgh, UK. [2] Selcia Ltd., Fyfield Business and Research Park, Fyfield Road, Ongar, Essex, UK. [3] Cambridge MedChem Consulting, Cambridge, UK. [4] Centre for Cardiovascular Science, College of Medicine and Veterinary Medicine, University of Edinburgh, Edinburgh, UK. [5] York Biomedical Research Institute, Hull York Medical School, University of York, York, UK. [6] Institute of Infection Immunity and Inflammation, College of Medical Veterinary Life-Sciences, University of Glasgow, Glasgow, UK. [7] York Biomedical Research Institute, Department of Biology, University of York, York, UK. [8] These authors contributed equally: Iain W. McNae, James Kinkead, Divya Malik. ✉email: simon.pettit@selcia.com; m.walkinshaw@ed.ac.uk

Human African trypanosomiasis (HAT), also known as sleeping sickness is caused by two subspecies of the parasitic protist *Trypanosoma brucei* and is transmitted by the bite of an infected tsetse fly[1]. The subspecies *T. b. gambiense*, prevalent in west and central Africa is responsible for over 97% of cases with the remainder caused by *T. b. rhodesiense* in eastern and southern Africa[2,3]. HAT has two stages: the first, haemolymphatic stage includes non-specific symptoms, such as headache and bouts of fever. The second stage occurs when the parasite has invaded the central nervous system (CNS), leading to progressive mental deterioration and ultimately death. There are five registered drugs currently used to treat HAT; all have a number of drawbacks, including severe side effects associated with significant toxicity/mortality or prolonged and complex dosing regimens, including a requirement for intravenous administration[4]. A new oral drug, fexinidazole has recently been given approval for use in the clinic (https://www.dndi.org/diseases-projects/portfolio/).

The bloodstream form (BSF) of *T. brucei* has evolved to rely on the high (5 mM) levels of glucose available in host blood as fuel. In this stage of its life cycle, the parasite mitochondrion is highly compromised and cannot carry out oxidative phosphorylation and *T. brucei* exclusively uses glycolysis as the sole source of ATP. Our hypothesis was therefore that the glycolytic pathway would be a suitable target for small molecule anti-HAT drugs. As proof of concept, RNA interference-mediated knockdown experiments showed that even a 50% decrease in glycolytic flux is sufficient to kill the parasite in vitro[5]. *T. brucei* phosphofructokinase (TbPFK) is located in peroxisome-related organelles called glycosomes[6] and carries out the third step in the glycolytic pathway, phosphorylating fructose 6-phosphate (F6P) to give fructose 1,6-bisphosphate (F16BP) (Fig. 1). Low sequence identity of ~20% with the three human isoforms (hPFK-M, hPFK-L and hPFK-P) despite sharing very similar active sites[7] supported the choice of this target.

The first published example of a TbPFK inhibitor was a dichlorophenyl-furanose derivative, which showed weak (IC$_{50}$ = 23 μM) inhibition[8]. A subsequent high-throughput screen (HTS) against TbPFK using a library of over 330,000 compounds[9] from the Molecular Libraries Small Molecule Repository, identified over 500 active compounds (https://pubchem.ncbi.nlm.nih.gov/bioassay/485367). The most potent compounds in this screen all contained a dichlorophenyl group. A convincing structure–activity relationship (SAR) showing the importance of the di-substituted phenyl group was developed and a series of compounds was synthesised, using an amidosulfonamide scaffold linked to the substituted phenyl group[10]. One of the best compounds in this series (CTCB-001, Fig. 2) had an IC$_{50}$ value of ~200 nM. This series of compounds also inhibited *Trypanosoma cruzi* PFK and showed up to fivefold better potency compared with *Tb*PFK. However, the compounds were not effective at killing either the *T. cruzi* or *T. brucei* parasites in in vitro culture with poor EC$_{50}$ values of at best ~20 μM, presumably due to poor uptake by the parasites[10].

In this study, we show how an X-ray structure-based design approach has led from an initial high-throughput screening hit to the synthesis of a series of a highly specific inhibitors against the trypanosome PFK that block glycolysis in the parasite, but have no effect on the human enzyme. Enzymatic and kinetic studies further show that the mechanism of action is allosteric and the inhibitors do not compete with ATP in the active site. Consistent with the mechanism, we show that the inhibitors have a very fast parasite killing time. The lead compounds have good pharmacokinetic (PK) profiles with excellent bioavailability. Studies with trypanosome-infected mice show the compounds readily penetrate the brain and that oral dosing can cure acute infection.

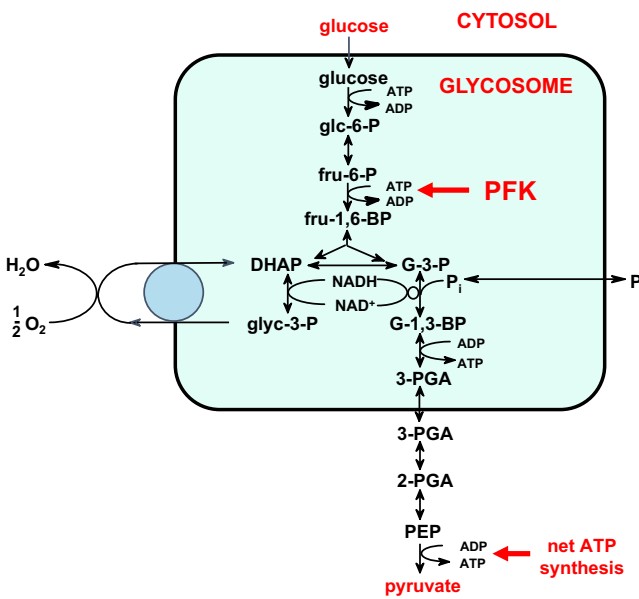

**Fig. 1 Glycolysis in bloodstream form *T. brucei*.** The first seven enzymes of the glycolytic pathway, from hexokinase to phosphoglycerate kinase, are sequestered in peroxisome-related organelles called glycosomes. Glucose, taken up by the trypanosome from the blood, enters the glycosomes and is converted stepwise via glucose 6-phosphate (glc-6-P), fructose 6-phosphate (fru-6-P), fructose 1,6-bisphosphate (fru-1,6-BP), glyceraldehyde 3-phosphate (G-3-P) and glycerate 1,3-bisphosphate (G-1,3-BP), which is the precursor of 3-phosphoglycerate (3-PGA) that exits the organelles. The last three steps occur in the cytosol with the conversion of 3-PGA to 2-PGA, the precursor of phosphoenolpyruvate (PEP) which is converted to pyruvate. Inside glycosomes, the use of ATP by hexokinase and PFK, and its formation by phosphoglycerate kinase are balanced. Net ATP synthesis by pyruvate kinase occurs in the cytosol. NADH formed inside the glycosomes by glyceraldehyde-3-phosphate dehydrogenase is re-oxidised by a glycosomal NADH-dependent glycerol-3-phosphate dehydrogenase, an electron shuttle involving glycerol 3-phosphate (glyc-3-P) and dihydroxyacetone phosphate (DHAP), and a mitochondrial FAD-dependent glycerol-3-phosphate oxidase system not coupled to ATP synthesis. The trypanosomes can also take up glycerol, to be converted into pyruvate with a net ATP synthesis of 1 ATP/glycerol, or be used in gluconeogenesis to form glucose 6-phosphate.

## Results and discussion

Bioactive lead compounds were developed using a structure-based approach. We used the SAR results of our previous studies[10] to select a small number of readily available fragment-like molecules that contained the dichlorophenyl pharmacophore. Each compound was tested in vitro in an enzyme inhibition assay and in a parasite killing assay (see 'Methods' and Supplementary Fig. 1) leading to the identification of CTCB-12, which potently inhibited TbPFK with an IC$_{50}$ value of 0.83 μM and high ligand efficiency of 0.51 (Fig. 2). It was also possible to co-crystallise CTCB-12 with TbPFK. The X-ray structure of this complex shows that the di-substituted phenyl group fills a deep allosteric pocket adjacent to the active site (Fig. 3), and explains the ubiquitous presence of the dichlorophenyl pharmacophore in hits from the HTS run. Based on the X-ray structure of CTCB-12, a number of derivatives were synthesised using pyrazole and bicyclic scaffolds, including imidazopyridines, pyrazolopyridines, thienopyridines and pyrrolopyridines, and tested against an enzyme inhibition assay and an in vitro parasite killing assay (Fig. 4). Over 30 X-ray structures of TbPFK complexed with ligands from most of the different scaffold families have been determined, and all show the conserved binding of the

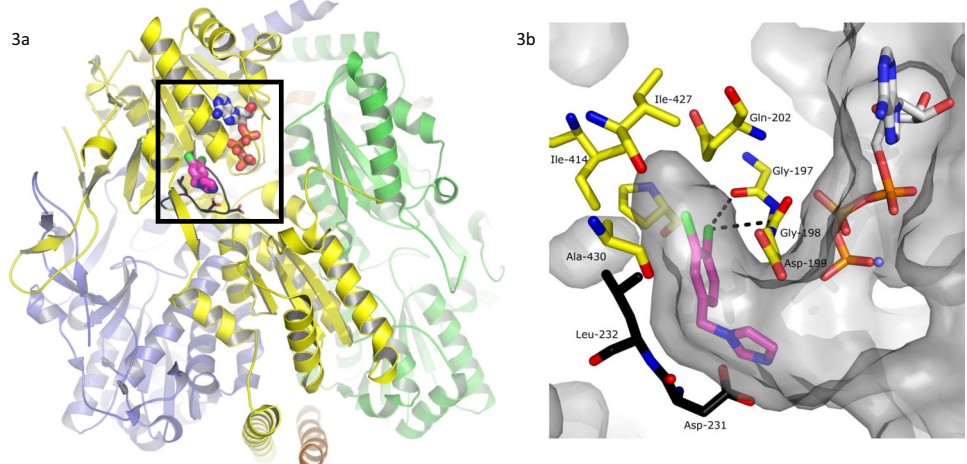

**2a**

| ID | Structure | IC$_{50}$ (μM) | EC$_{50}$ (μM) | L.E. | Calc pKa |
|---|---|---|---|---|---|
| CTCB001 | | 0.22±0.03 | 20.2±2.1 | 0.29 | -2 |
| CTCB012 | | 0.83±0.01 | 8.9±0.2 | 0.51 | 6.47 |
| CTCB107 | | 1.14±0.76 | 8.5±0.6 | 0.39 | 5.19 |
| CTCB103 | | 0.46±0.03 | 7.8±0.1 | 0.41 | 7.84 |

**2b**

| ID | R$^1$ | R$^2$ | R$^3$ | IC$_{50}$ (μM) | EC$_{50}$ (μM) | L.E. | Calc pKa |
|---|---|---|---|---|---|---|---|
| CTCB360 | H | Cl | Cl | 0.07±0.03 | 1.1±0.1 | 0.44 | -5.48 |
| CTCB401 | HO | Cl | Cl | 0.07±0.03 | 0.91±0.04 | 0.38 | -2.55 |
| CTCB421 | H$_2$N | Cl | Cl | 0.03±0.02 | 0.54±0.09 | 0.40 | 8.99 |
| CTCB435 | H N | Cl | Cl | 0.12±0.01 | 0.25±0.05 | 0.35 | 9.29 |
| CTCB405 | N | Cl | Cl | 0.18±0.03 | 0.37±0.03 | 0.33 | 8.32 |
| CTCB470 | N | Br | Cl | 0.22±0.08 | 0.31±0.03 | 0.32 | 8.32 |
| CTCB508 | N | Br | F | 0.20±0.08 | 0.30±0.03 | 0.33 | 8.32 |
| CTCB457 | H$_2$N | Cl | Cl | 0.11± 0.02 | 0.27±0.01 | 0.35 | 9.02 |
| CTCB481 | H N | Cl | Cl | 0.12±0.07 | 0.15±0.05 | 0.34 | 9.43 |

**Fig. 2 Optimisation of the CTCB series of inhibitors of TbPFK.** IC$_{50}$ values (μM) for inhibition of *T. brucei* phosphofructokinase. EC$_{50}$ values (μM) for in vitro parasite killing of the bloodstream form of *T. brucei* strain Lister 427 (see Supplementary Methods 2.1 and 4.2). L.E. ligand efficiency. IC$_{50}$ values are based on at least three independent measurements ('biological replicates'). EC$_{50}$ values were initially determined from two technical replicates. Estimated standard deviations (ESDs) for selected compounds were determined using biological replicate studies ($n \geq 3$). The pK$_a$ values were calculated using ChemAxon software and categorised assuming physiological pH of 7.8 (Supplementary Table 6). The figure shows that there is a better translation from IC$_{50}$ to EC$_{50}$ for the more basic molecules. **a** The path to identifying the chemical scaffold used for lead compound development CTCB-001, the starting point for this work, was identified from a high-throughput screen as a good enzyme inhibitor but with poor trypanocidal activity[10]. **b** Inhibition data for derivatives of the pyrrolopyridine series. Details of the synthesis and characterisation of CTCB-405, CTCB-470 and CTCB-508 are given in Supplementary Methods 7 and 8. Source data for IC$_{50}$ and EC$_{50}$ values are provided as a Source Data File.

**Fig. 3 Architecture of the TbPFK target and X-ray structure of the complex with CTCB-12. a** Tetrameric structure of TbPFK. The four TbPFK chains making the TbPFK tetramer are coloured (yellow, green, orange and purple). The boxed region shows the position of the active site with ATP modelled in position (carbon atoms shown as white-coloured thick sticks). The fragment CTCB-12 is in the adjacent allosteric pocket shown in pink, chlorine atoms are green). Part of the mobile activating loop (residues 225PKTIDNDLSFS235) is highlighted in black. **b** Blow up of the boxed region showing the dichlorophenyl binding pocket in TbPFK complexed with fragment structure CTCB-12 with the dichlorophenyl ring filling the allosteric pocket. ATP is also shown coordinated to magnesium (blue sphere). Two residues from the mobile activating loop (L232 and D231) are in black. The Supplementary Movie 1 shows how the CTCB compounds block activation and prevent the side chain of L232 moving into the allosteric pocket, stopping the mobile activating loop adopting the active R-state conformation, which brings D231 and D229 within coordinating distance of the substrate molecules. Supplementary Figure 7 shows stereo diagrams with electron density contoured round CTCB-12, CTCB-360 and CTCB-405.

dichlorophenyl group locked in the allosteric pocket as observed in the structure with CTCB-12.

The most efficacious group of compounds were derivatives of a pyrrolopyridine scaffold with amino side chains on N5 (Fig. 5). Ethylamino substituents were found to have the best inhibitory activities, with IC$_{50}$ values as low as 30 nM (CTCB-421, Fig. 2). X-ray structures of this series show that the amino side chain substituent on N5 bends to make a hydrogen bond with Asp199

(Fig. 5b). This binding mode is further stabilised by a short (3.1 Å) electrostatic interaction between the negative carboxyl oxygen and the CTCB-405 amide carbon atom. The X-ray complexes clearly explain the SAR that strictly limits the size of substituents on the phenyl ring (Fig. 5a). Chlorine in the meta position is ideally positioned to make a specific electrostatic interaction with a carbonyl oxygen of Gly197 lining the allosteric pocket (Fig. 5b). The geometry of this 'halogen bond' with a Cl…O distance of 3.3 Å

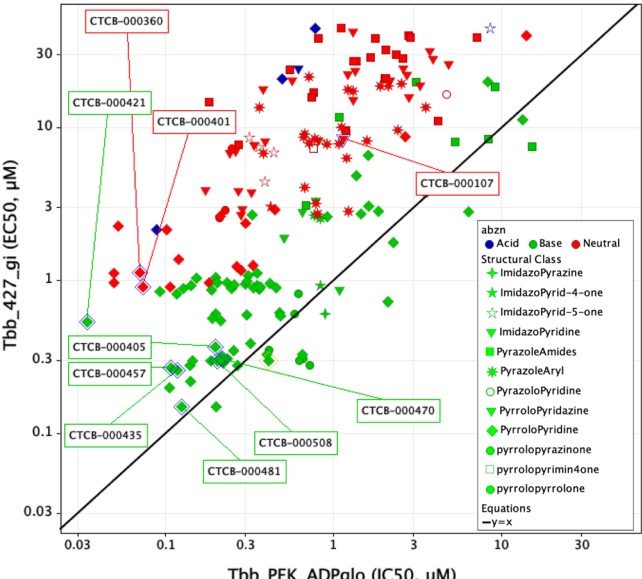

**Fig. 4 EC$_{50}$ versus IC$_{50}$ values for CTCB compounds.** Each compound in the CTCB series was tested against *T. brucei* in an in vitro killing assay and also tested in an enzyme inhibition assay against TbPFK. The *Y*-axis-labelled Tbb_427_gi gives the EC$_{50}$ values for the growth inhibition assay (Supplementary Methods 4) and the *X*-axis-labelled Tbb_PFK_ADPglo gives IC$_{50}$ values for the enzyme inhibition assay (Supplementary Methods 2.1). The compounds are colour coded according to charge (green = basic, red = neutral and blue = acidic). The plot shows that for a wide variety of structural classes (denoted by shape of the data point), there is a better translation from IC$_{50}$ to EC$_{50}$ for the basic molecules and that the effect of pK$_a$ on parasite killing appears to be consistent across different structural classes (though only details of the pyrrolopyridine class are described in this communication). The pK$_a$ values were calculated using ChemAxon software and categorised assuming physiological pH of 7.8.

and C–Cl–O angle of 150° is in the expected range[11]. Other permutations of Br and F substituents on the phenyl group were found to have a small effect on binding affinity, but showed significant effects on PK behaviour.

Isothermal titration calorimetry (ITC) results show that all of the CTCB compounds bind with an almost perfect 1:1 stoichiometry with TbPFK (Supplementary Fig. 5). The main binding contribution for the series is enthalpic with a relatively minor unfavourable entropic contribution. For CTCB-405, the $\Delta G$ of $-9.57$ kcal mol$^{-1}$ comprises $\Delta H = -13.6$ kcal mol$^{-1}$ with a $T\Delta S$ contribution of $-4.03$ kcal mol$^{-1}$. The $K_d$ values match well with the surface plasmon resonance (SPR) results; CTCB-405 has a $K_d$ of 92 nM (ITC) compared with 82 nM (SPR). SPR data (Supplementary Fig. 6) also show that the CTCB compounds have fast on rates ($k_{on}$CTCB-405 $= 0.96 \mu$M$^{-1}$ s$^{-1}$) and slow off rates ($k_{off} = 0.079$ s$^{-1}$).

The compounds in Fig. 4 are coded to show the variety of chemical scaffolds that have been synthesised and tested for both parasite killing and TbPFK inhibition (further details of synthetic routes and chemical characterisation for many of these TbPFK inhibitors are available in a published patent application[12]). The parasite killing assay uses the fluorescent signal from the redox indicator Alamar Blue as a measure of cell viability, and the generally good correlation between IC$_{50}$ versus EC$_{50}$ fits with TbPFK being the target for the CTCB compound series. There is little evidence for off-target effects in which the EC$_{50}$ killing concentration could be expected to be less than the IC$_{50}$ enzyme inhibition concentration. Deviation from a direct correlation (where EC$_{50}$ > IC$_{50}$) can be rationalised by poor uptake of the

compound into the parasites or into the glycosome organelles that contain TbPFK. The series of basic molecules (coloured green in Fig. 4) show good correlation of IC$_{50}$ (PFK inhibition) with EC$_{50}$ (parasite killing). However, neutral compounds (shown in red) or acidic compounds (shown in blue) have significantly poorer EC$_{50}$ values. The effect of pK$_a$ on parasite killing seems to be consistent for a wide variety of structural classes. The converse observation that small chemical changes affecting pK$_a$ can cause significant changes in EC$_{50}$ values also supports the idea that basicity plays an important role in uptake and killing. For example, when the dimethylamino group of compound CTCB-405 is replaced by a hydroxyl group (CTCB-401, Fig. 2), the neutral compound CTCB-401 inhibits the enzyme more effectively with an IC$_{50}$ of 73 nM (compared with 180 nM for CTCB-405), while it is less effective against the parasite with an EC$_{50}$ of 900 nM compared with 370 nM for the more basic CTCB-405. X-ray structures of both compounds show the same binding pose. As shown in Figs. 2 and 4, all of the more potent compounds have pK$_a$ >8 and include primary amines (e.g., CTCB-421 and CTCB-457) and secondary amines (e.g., CTCB-435 and CTCB-481). Despite the potency against the parasites, subsequent ADME and toxicity studies showed these generally more basic compounds had a less suitable profile than the tertiary amine lead compounds CTCB-405, CTCB-470 and CTCB-508 which, though not having the lowest IC$_{50}$ or EC$_{50}$ values, were found to have the best combination of in vitro and in vivo properties. Importantly, the lead compounds also show good activity against the human pathogens *T. b. gambiense* and *T. b. rhodesiense* with EC$_{50}$ values between 150 and 250 nM, showing even slightly better potency than against the *T. b. brucei* Tb427 and TbGVR35 laboratory strains (Supplementary Table 4).

Structural, enzymatic and binding studies of TbPFK characterise the inhibitory mechanism. Trypanosomatid PFKs have been characterised by X-ray crystallography and two conformational states have been identified that fit with a classical description of an allosteric enzyme that transitions from an inactive T-state conformation to an active R-state conformation[13]. Comparison of the T-state and R-state structures show that activation of TbPFK requires a large movement of the critical catalytic residues Asp229 and Asp231 (Fig. 3 and Supplementary Movie 1). The carboxyl groups of the two Asp residues hydrogen bond with the F6P substrate and also coordinate a catalytic magnesium ion, facilitating transfer of a phosphoryl group from ATP. The mobile activating loop is locked in its active R-state by the side chain of Leu232, which sits on the same mobile loop and fits into the allosteric drug-binding pocket (Fig. 3 and Supplementary Movie 1). The mode of action for the CTCB family of inhibitors is to lock the tetramer in the inactive T-state, with the activation loop held remote from the substrate molecules.

Enzyme kinetic studies confirmed that the CTCB compounds are not competitive against either ATP or F6P. TbPFK inhibition was studied using an enzyme assay, in which production of ADP by TbPFK was coupled to the reactions of pyruvate kinase and lactate dehydrogenase: the conversion of pyruvate to lactate and NADH to NAD$^+$, is monitored by reduction of UV absorbance at 340 nm. The Michaelis–Menten plots (Supplementary Fig. 2 and Supplementary Table 1) show inhibitory behaviour for the compound CTCB-405, which is typical for the compound series. The reduction of $V$max for both substrates (F6P and ATP) indicates that the inhibitor is not competitive with either ATP or F6P. This is consistent with the X-ray structures showing no steric overlap between the substrate binding sites and the CTCB inhibitor binding site.

Divergent allosteric mechanisms of hPFK and TbPFK explain the species specificity of the CTCB compounds. Of the 19 residues that bind the ATP and F6P substrate molecules, 12 are

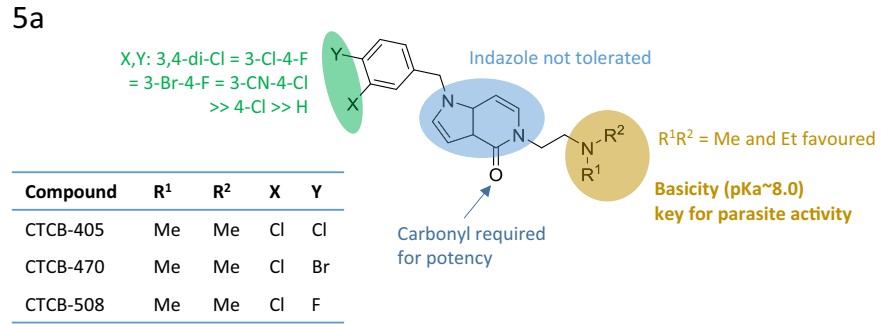

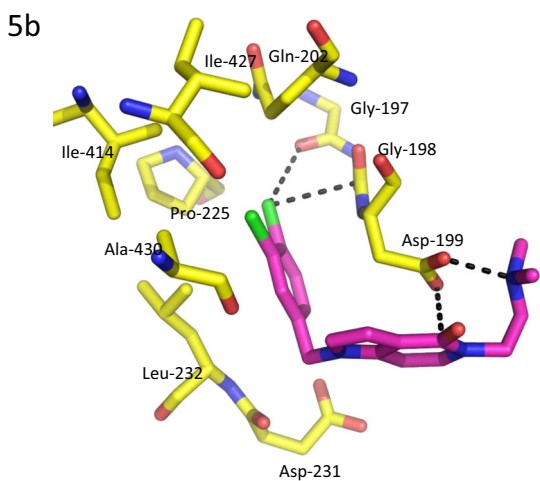

**Fig. 5 Structure–activity relationship of the CTCB allosteric inhibitors and binding interaction of the lead CTCB series. a** Structure–activity of the pyrrolopyridine scaffold. Three regions in the structure are highlighted in green, blue and yellow, and the respective biological properties or activities are indicated. **b** X-ray structure of CTCB-405 bound to TbPFK showing three binding features of the pyrrolopyridine lead series. The chlorine atom forms a 'halogen bond' with the backbone carbonyl atom from Gly (Cl...O = 3.3 Å). There is an unusual short attractive interaction between the carboxyl oxygen atom of Asp199 and the amide carbon atom of CTCB-405 (O...C = 3.1 Å). The ethylamine side chain adopts a conformation allowing salt bridge formation with Asp199 (N...O = 2.6 Å).

conserved between human and TbPFK. However, the overall sequence identity between the two orthologues is only 20%. Human PFK isoforms are tetramers with chain length of ~780 amino acids and have additional regulatory domains responsible for a very different allosteric regulatory mechanism compared with the smaller 487 residues long TbPFK[14]. The allosteric pocket regulating the switch between T- and R-states in the TbPFK is not present in any of the hPFK isoforms (Supplementary Fig. 3 and Supplementary Movie 2). The inhibitory mechanism of the pyrrolopyridine compounds is therefore exquisitely specific for trypanosomatid PFK with no effect against any of the three hPFK isoforms even at concentrations above 100 µM (Fig. 6 and Supplementary Fig. 4).

The time-to-kill (TTK) assay shows very fast parasite killing. As predicted from the biology of the BSF that relies wholly on glycolysis for ATP production, inhibition of the pathway results in a very fast TTK. Two experimental methods were used to measure ATP levels as a real-time indicator of viability. For Tb427 parasites, a Cell Titre Glo kit was used and for the TbGVR35 bioluminescent strain, killing was assessed by adding luciferin which requires ATP to produce a luminescent signal (Supplementary Methods 4.3 and 4.4). Both assays show that for the lead CTCB series, over 99% of parasites are killed in under 30 min at a concentration of 4 µM (Fig. 6b). The BSF of *T. brucei* cannot produce ATP by oxidative phosphorylation and is solely dependent on flux through the glycolytic pathway. The clear

correlation between inhibitor dose and reduction in ATP production, as observed in the TTK assays strongly supports the proposed mode of action of the CTCB compounds is by blocking glycolysis. It is a long-established experimental observation that withdrawal of glucose nutrient leads to rapid loss of motility and cell death[15]. It has recently been shown that the BSF of *T. brucei* can proliferate on high levels of glycerol; however, we have shown that addition of glycerol to the medium has little effect on parasite killing by the CTCB compounds (Supplementary Table 3). The very fast ($t_{0.5}$ < 15 min) parasite killing linked to loss of ATP production (Fig. 6b) caused by the CTCB compounds is therefore wholly consistent with the direct inhibition of one of the glycolytic enzymes. This mode of action resulted in a much faster kill than any of the other trypanocidal compounds tested in our assay, including fexinidazole, suramin and those of the oxaborole family, which all have TTK values >20 h.

In order to estimate a killing-dose for in vivo trials, a series of washout experiments were carried out: parasites were incubated with a given concentration of compound before washing out the drug and determining whether the parasites were still viable and able to proliferate. The results showed that even though >99% of parasites were killed within 30 min, a longer exposure of between 2 and 4 h at concentrations of 2–4 µM was required to ensure 100% killing in vitro (Supplementary Fig. 8). The existence of a small population of parasites more resistant to the effects of the CTCB compounds might be explained by subpopulations of

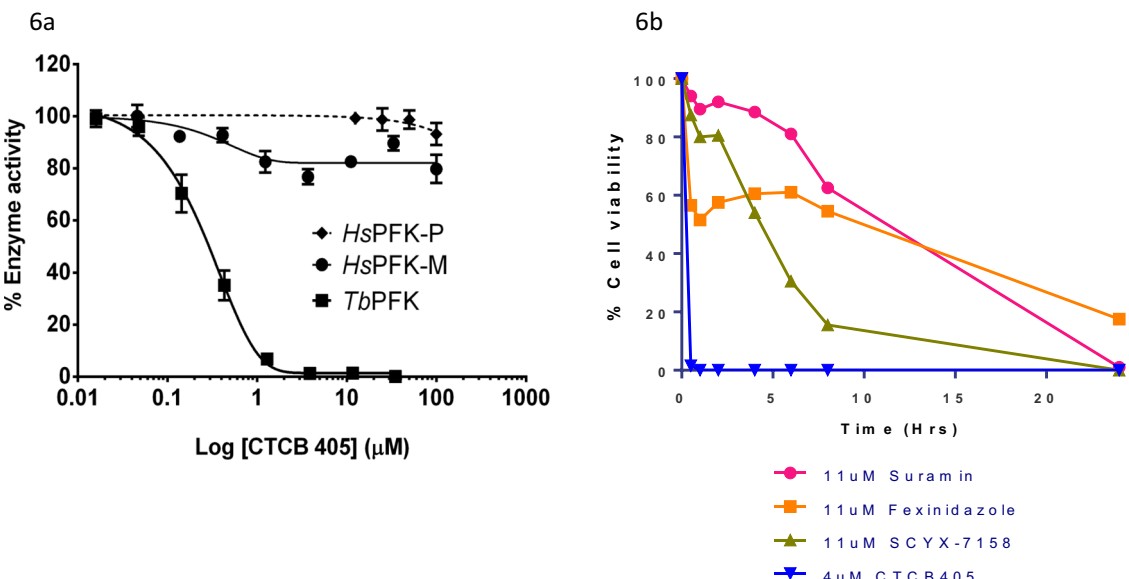

**Fig. 6 CTCB compounds are highly specific and have a very fast time-to-kill compared with all available clinical anti-HAT drugs. a** CTCB-405 shows no significant inhibition of human PFK. Enzyme activity was measured using an aldolase/TIM linked assay at 25 °C. Enzyme activities of *T. brucei* PFK, hPFK-M and hPFK-P were measured as described in Supplementary Methods 2.3 (Supplementary Fig. 4 shows similar result for titrations of CTCB-508, and CTCB-470 against hPFK-L). Experiments were carried out in duplicate. Data are presented as mean values ± standard deviation. Highest concentration of CTCB-405 = 100 μM. Source data are provided as a Source Data File. **b** Kill time of CTCB-405 compared with existing HAT drugs in the clinic. The cell viability assay was used to study the effect of a fixed concentration of suramin (11 μM), fexinidazole (11 μM), SCYX-7158 (11 μM) and CTCB-405 (4 μM). Over 99% of trypanosomes were killed in under 30 min compared with much slower killing rates for the other drugs. The points on the graph for CTCB-405 were determined from two independent biological replicate experiments with each time point measured twice (using duplicate technical replicates). The ESD for each point on the graph is between 0.7 and 1.4%. For drugs in clinical use each point on the graph is the average of two technical replicates from one experiment (with a maximum difference of 7.8% for any of the points). Source data are provided as a Source Data File.

**Table 1 In vitro ADME profiles for CTCB-405, CTCB-470 and CTCB-508. Methods are described in Supplementary Methods 5 and 6. Enzyme assay results are means ± SD in multiple assays. ADME results are averages of $n = 2$ or $3$ measurements in individual assays. $MLM$ $Cl_{int}$ mouse liver microsome intrinsic clearance, $HLM$ $Cl_{int}$ human liver microsome intrinsic clearance.**

| Parameter | CTCB-405 | CTCB-470 | CTCB-508 |
|---|---|---|---|
| PFK IC$_{50}$ (μM) ($n = 4$) | 0.18 ± 0.03 | 0.22 ± 0.08 | 0.20 ± 0.05 |
| Tbb EC$_{50}$ (μM) ($n = 2$) | 0.37 ± 0.04 | 0.31 ± 0.03 | 0.30 ± 0.02 |
| ADME results | | | |
| MLM Cl$_{int}$ (μL min$^{-1}$ mg$^{-1}$) | 70 | 91 | 56 |
| HLM Cl$_{int}$ (μL min$^{-1}$ mg$^{-1}$) | <8.6 | 17.0 | <8.6 |
| Human hepatocytes Cl$_{int}$ (μL min$^{-1}$ per 10$^6$ cells) | 2.3 | nd | nd |
| Mouse hepatocytes Cl$_{int}$ (μL min$^{-1}$ per 10$^6$ cells) | 3.7 | 3.6 | nd |
| HepG2 EC$_{50}$ (μM) | 20.8 | 12.3 | 33.4 |
| Selectivity | 56 | 39 | 111 |
| MDCK-MDR1 A-B/B-A | 17.6/17.2 | 34.9/28.3 | — |
| CYP450 2C9 (% inhibition at 10 μM) | 11 | 28.5 | 6.1 |
| Human hERG IC$_{50}$ (μM) | 3.98 | 1.69 | 4.75 |
| Human plasma protein binding $f_u$(%) | 6.3 | 3.2 | 21 |
| Mouse plasma protein binding $f_u$(%) | 5.8 | 6.0 | 18 |
| Mouse brain tissue binding $f_u$ (%) | 0.8 | 0.6 | 1.7 |
| $K_{p,brain}$ | 4.1 | 1.5 | 3.0 |
| $K_{p,uu,brain}$ | 0.6 | 0.2 | 0.3 |

Brain and plasma concentrations were measured 2 h after oral doses of 50 mg kg$^{-1}$ (CTCB-405 and CTCB-470) and 100 mg kg$^{-1}$ (CTCB-508).

parasites being present at different stages of the cell cycle. The parasite cultures used in the killing experiments are not synchronised, and it is likely that ATP requirement varies depending on cell cycle phase and a subpopulation with lower ATP requirement may be less susceptible to the CTCB compounds, however, this phenomenon needs further investigation.

The CTCB lead compounds have good ADME and pharmacokinetic (PK) profiles (Tables 1 and 2). Potential compound toxicity

was determined by measuring the ratio of EC$_{50}$ values for human HepG2 cells compared with the EC$_{50}$ for parasite killing giving values of between 40 and 110-fold selectivity (Tables 1 and 2). CTCB-405 was very stable when tested in human liver microsomes, but was much less stable when tested in mouse liver microsomes (Cl$_{int}$ < 8.6 compared with 70 μL min$^{-1}$ mg$^{-1}$). MDR1-MDCK permeability assays showed that CTCB compounds are not P-gp substrates and cross the endothelial membrane.

**Table 2 Mouse pharmacokinetic data for CTCB-405, CTCB-470 and CTCB-508.**

|  | CTCB-405 | | CTCB-470 | | CTCB-508 | |
|---|---|---|---|---|---|---|
| Route | IV | PO | IV | PO | IV | PO |
| Dose (mg kg$^{-1}$) | 1 | 5 | 1 | 10 | 1 | 10 |
| Elimination $t_{1/2}$ (h) | 2.7 | | 3.1 | | 0.7 | |
| AUC (infinity) (ng-h mL$^{-1}$) | 260 | 704 | 657 | 1565 | 349 | 4170 |
| Cmax (ng mL$^{-1}$) | | 206 | | 393 | | 1333 |
| Tmax (h) | | 0.3 | | 0.5 | | 2.0 |
| $V_d$ (area) (L kg$^{-1}$) | 12.3 | | 6.9 | | 2.9 | |
| CL/kg (L h$^{-1}$ kg$^{-1}$) | 3.8 | | 1.5 | | 2.7 | |
| Bioavailability (F%) | | 54 | | 24 | | 119 |

Pharmacokinetic parameters of the compounds in plasma were calculated from the curves using PKSolutions software. Compounds were administered either intravenously or orally at the doses indicated ($n = 1$ for each compound). 'Methods' are described in Supplementary Methods 6.1 and Supplementary Fig. 9).

Female CD-1 mice were used to establish the tolerance and efficacy of compounds in vivo (Supplementary Methods 6).

PK parameters for the lead compounds were determined using three mice per data point (Supplementary Methods 6.1). The nature of the dihalogen substituents was found to have a significant effect on oral bioavailability with values for CTCB-508, CTCB-405 and CTCB-470 of 119%, 54% and 24%, respectively (Table 2). CTCB-508 also has a relatively low mouse plasma protein binding value of 82% compared with CTCB-405 (94%) and CTCB-470 (94.2%).

The PK studies also show that the CTCB series can cross the blood–brain barrier and enter the brain. A single oral dose of 50 mg kg$^{-1}$ for CTCB-405 gives a mean plasma concentration of 366 ng mL$^{-1}$ and a mean concentration in brain homogenate of 1508 ng g$^{-1}$ resulting in a $K_p$ value (ratio of total compound in brain to total compound in plasma) of 4.1. Binding of CTCB-405 to brain tissue is about sevenfold greater than to plasma with free fraction brain = 0.8% compared with free fraction plasma = 5.8% (Table 1). These values for concentrations of free compound in the brain give a $K_{p,uu}$ value of 0.6 (defined as the ratio of unbound compound in brain to unbound compound in plasma). This ratio provides an estimate of the amount of free drug in the brain interstitial fluid as being just over half the free concentration in the blood.

High doses of each of the three lead compounds, CTCB-405, CTCB-470 and CTCB-508, were well tolerated. Mice treated with CTCB-405 at up to 300 mg kg$^{-1}$ per day in cumulative dosing tolerance studies showed no apparent toxicity and autopsied livers of mice subjected to such high doses appeared normal. A dosing regimen with three doses of 50 mg kg$^{-1}$ given every 2 h resulted in a plasma concentration of $6.8 \pm 2.1\,\mu$M measured 2 h after the final dose. A similar regimen with three doses of 100 mg kg$^{-1}$ gave a plasma concentration of $9.8 \pm 3.1\,\mu$M again measured 2 h after the final dose. The predicted plasma values based on the measured mouse PK parameters were 4.1 and $8.3\,\mu$M, respectively (Supplementary Fig. 10 and Supplementary Table 5).

*T. b. brucei* Lister 427 infection is cured by 1-day dosing. The majority of the compounds in the literature have been tested in mice infected with the *T. b. brucei* Lister 427 strain, which is a monomorphic (non-pleiomorphic) strain that grows in a virulent and unconstrained manner and does not undergo life cycle differentiation. These parasites have a doubling time of ~6 h [16] and infection with $1 \times 10^5$ parasites usually requires euthanasia of mice within 4–5 days of infection in the absence of drug

treatment. Drug efficacy is determined by taking regular blood samples from infected mice and checking microscopically for parasitaemia. In this model of acute infection a single surviving parasite should proliferate to reach detectable levels in ~5 days and a 30-day period of no parasite detection is considered as cure.

In vitro parasite killing experiments described above suggested that in order to kill 100% of the parasites, an exposure to over $2\,\mu$M compound would be required for a period of 8 h or alternatively exposure to a higher total concentration of compound of $4\,\mu$M for a shorter (6 h) period (Supplementary Fig. 8). A simple model based on estimated bioavailability and the measured half-life of the compound in plasma showed that these levels could be achieved by three doses of 100 mg kg$^{-1}$ administered at 2 h intervals. Experimentally, three doses of CTCB-405 at 100 mg kg$^{-1}$ given on two consecutive days gave plasma levels of $9.8\,\mu$M (Supplementary Table 5).

A number of dosing regimens for the three lead compounds were tested against infected mice (Fig. 7a and Supplementary Fig. 11). Four single day doses of CTCB-405 at 50 mg kg$^{-1}$ delayed the onset of parasitaemia by up to 12 days; however, thrice daily doses of 50 mg kg$^{-1}$ gave a complete cure (defined as preventing the recurrence of parasitaemia for >30 days; Supplementary Fig. 11). A 1-day dosing regime was explored for CTCB-405 with three doses of 100 mg kg$^{-1}$ at 2 h intervals which also resulted in complete cure for 5/5 infected mice.

An alternative mouse model for stage 1 (acute) and stage 2 (CNS-involved) HAT has been developed using a transgenic cell line derived from the pleiomorphic *T. b. brucei* GVR35 that expresses red-shifted firefly luciferase[17,18]. EC$_{50}$ values of CTCB compounds for the bioluminescent *T. b. brucei* GVR35 strain determined in vitro were similar to those for *T. b. brucei* Lister 427. *T. b. brucei* GVR35-infected mice present stage 1 disease by day 7 post infection with progression to stage 2 by day 21. By injecting the mouse with luciferin, the level of parasitaemia can be estimated in real time by monitoring the luminescence. Unlike monomorphic Lister 427 *T. b. brucei*, the pleiomorphic GVR35 *T. b. brucei* is controlled by a quorum-sensing mechanism resulting in the significant increase in survival time in mice and lower blood parasitaemia that occurs in waves. This extended survival time is also accompanied by parasite dissemination to multiple tissues and organs, including the skin, adipose tissue, lung and brain, with parasite loads increasing over time[17,19]. This more virulent strain usually requires higher or longer doses of trypanocides for cure in vivo compared to the Lister 427 strain[20,21], possibly caused by reduced drug accessibility in these extravascular tissues. Using this more stringent model, treatment with CTCB-405 during stage 1 (day 7) reduced parasitaemia in the body and blood of mice, and delayed detection of parasites in blood to day 14 post infection. Clearance of parasitaemia is not complete as seen with the Lister 427 monomorphic strain, but treatment in stage 2 infection (from day 21) with CTCB-470 and CTCB-508 showed fast and significant signal reduction in the body (Fig. 7b). Given that the dosing regimen in place would not result in cure of the GVR35-infected mice, the stage 2 experiment was terminated on day 23 and brains removed. The brains of saline-perfused mice also showed very clear reduction in parasitaemia (Fig. 7c, d, and Supplementary Figs. 12 and 13) confirming compound activity in the CNS, but without achieving sterile parasitological cure.

The CTCB family of compounds described here have good oral availability, cross the blood–brain barrier, are exquisitely species specific and show very rapid parasite killing. Though more development would be required before the CTCB compounds could be considered as clinical candidates, the encouraging in vivo results showing clearance of parasitaemia in the mouse suggest that trypanosome PFK provides a new and species-

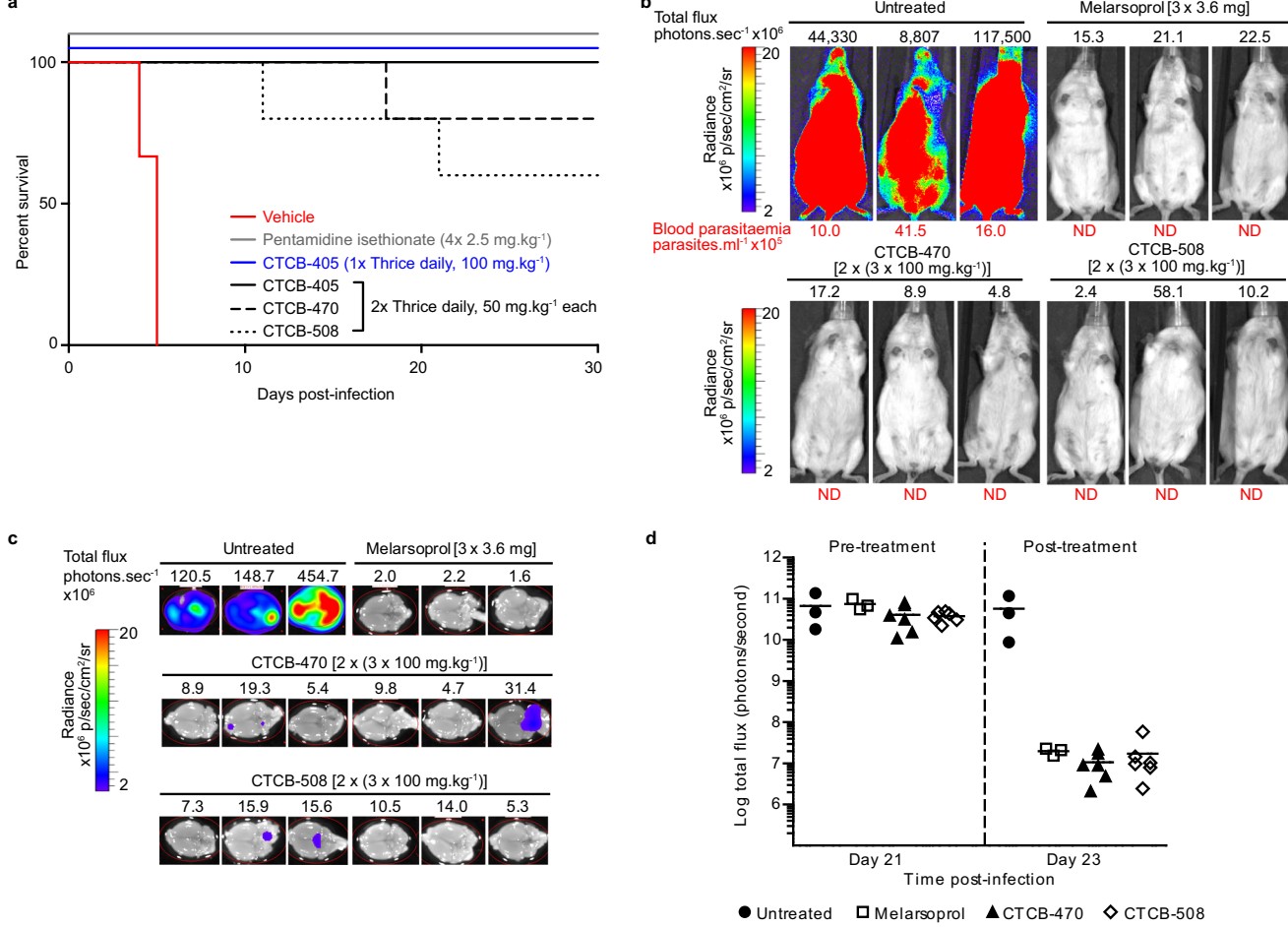

**Fig. 7 CTCB compounds clear parasites in a stage 1 model of HAT infection with 1-day oral dosing and show reduction of parasitaemia in the brain in a pleiomorphic model. a** Kaplan–Meier survival curve depicting cure in *T. b. brucei* Lister 427-infected mice treated with CTCB-405 (solid line, blue), with no recurrence of parasitaemia 30 days post infection. Three doses of CTCB-405 at 100 mg kg$^{-1}$ were administered 2 h apart on a single day ($n = 5$). The curves also demonstrate either complete clearance or a significant increase in survival of mice orally dosed thrice daily at 50 mg kg$^{-1}$ over 2 days ($n = 5$) with CTCB-405 (solid line, black), CTCB-470 (dashed line, black) or CTCB-508 (dotted line, black) ($p < 0.0001$). Control mice were administered either vehicle only ($n = 3$; solid line, red) or four single daily doses of 2.5 mg kg$^{-1}$ pentamidine isethionate intraperitoneally ($n = 3$; solid line, grey). Survival of mice was assessed for 30 days post infection. Mice were euthanised upon detection of first visible parasites in blood samples by microscopy. **b** Bioluminescence live imaging of *T. b. brucei* GVR35 after treatment (day 23) to evaluate parasitaemia in vivo. Infected mice were treated topically with three doses of 3.6 mg melarsoprol over 3 days ($n = 3$) or orally with six doses of CTCB compounds (CTCB-470 or CTCB-508) at 100 mg kg$^{-1}$ over 2 days ($n = 6$, only three mice from each batch are shown; Melarsoprol is a toxic anti-stage 2 HAT drug used as a positive control). Whole body bioluminescence (total flux in photons per second, above image) and blood parasitaemia (in parasites mL$^{-1}$, below image) for three representative mice from each group is shown. **c** At day 23 mice shown in **b** were perfused with saline; brains were removed, soaked in luciferin and imaged to detect bioluminescent parasites (bioluminescence of brains imaged ex vivo on day 23 quantified in Supplementary Fig. 12 and side head images of mice shown in Supplementary Fig. 13). **d** Whole body bioluminescence pre- (day 21) and post-treatment (day 23) of infected mice shown in **b** ($n = 3$ for untreated and melarsoprol-treated mice and $n = 6$ for both CTCB-470 and CTCB-508-treated mice).

specific drug target for parasite infection. The glycolytic pathway is one of the oldest and best-conserved cellular pathways across both eukaryotes and prokaryotes. This high degree of conservation along with the ubiquitous presence of phosphorylated substrates has made enzymes in this pathway unappealing and as yet unexploited as drug targets. However, though the active sites of most of the enzymes in the pathway have been conserved among different organisms, the allosteric mechanisms that regulate each step have evolved and diverged so that each organism can uniquely regulate and tune the pathway to suit their individual biological niche. This paper provides the first example of a family of drug-like molecules that specifically inhibit this pathway and opens up the possibility of developing more families of species-specific inhibitors. In particular, there are opportunities to explore the sophisticated allosteric regulatory mechanisms of the

human glycolytic enzymes that play an important role in cancer and inflammation or in the inhibition or Gram-positive and -negative bacteria, where the biological niches of pathogenic bacteria have led to major differences in allosteric regulatory mechanisms compared to enzymes of the host.

## Methods

**Enzymatic measurements**. Parasite enzyme kinetic data were measured using N-terminally His$_6$-tagged TbPFK expressed in *Escherichia coli* C41 cells, using a codon optimised gene inserted in a pET28a vector (Supplementary Methods 1). Recombinant human PFKs were expressed in PFK-deficient *Saccharomyces cerevisiae*[22]. PFK enzyme inhibition was measured using an ADP-Glo$^{TM}$ assay (Promega) in a 96-well plate format (Supplementary Methods 2.1), using a Spectramax© M5 Multimode Plate Reader to measure luminescence values which were converted to ADP concentrations. IC$_{50}$ values were determined using a sigmoidal curve fitting algorithm in Kaleidagraph 4 (Supplementary Fig. 1). Mode of

action studies were carried out using a linked enzyme assay coupling the forward reaction of TbPFK to the M1 isoform of human pyruvate kinase followed by lactate dehydrogenase. The change in UV absorbance at 340 nm resulting from the conversion of NADH to $NAD^+$ was measured at 25 °C in 'kinetic mode' on a Spectramax© M5 Multimode plate reader. Initial velocities were plotted against substrate concentration to give curves showing non-competitive inhibition (Supplementary Fig. 2). An alternative linked enzyme assay was developed to control for the potential inhibitory activity of ATP on human PFK (Supplementary Methods 2.3).

**Biophysical binding studies**. ITC experiments were carried out in HBS buffer (10 mM HEPES, 150 mM NaCl, 0.005% surfactant p20, pH 7.4, 1% DMSO). Protein stocks were desalted (HiTrap 5 mL desalting column, GE Healthcare) with the above buffer to ensure minimal buffer mismatch between ligand and analyte. Compound titrations consisted of 16 injections with 180 s delay between injections and 750 r.p.m. stirring at 25 °C. All solutions were degassed prior to use. A typical ITC trace showing 1:1 binding stoichiometry is shown in Supplementary Fig. 5.

SPR measurements were performed using a BIAcore T200 instrument (GE Healthcare). Active N-terminally His$_6$-tagged TbPFK surfaces were generated using a His-tag capture and coupling onto $Ni^{2+}$-nitrilotriacetic acid sensor chips. Following $Ni^{2+}$ priming (30 s injection of 500 μM NiCl$_2$ at 5 μl min$^{-1}$), dextran surface carboxylate groups were minimally activated by an injection of 0.2 M EDC; 50 mM NHS at 5 μl min$^{-1}$ for 240 s. TbPFK (at concentrations between 10 and 400 nM), in 10 mM NaH$_2$PO$_4$, pH 7.5; 150 mM NaCl; 50 μM EDTA; 0.05% surfactant p20, was captured via the hexa-His-tag and simultaneously covalently stabilised to between 1500 RU and 3100 RU, by varying the contact time. Immediately following the capture/stabilisation a single 15 s injection of 350 mM EDTA and 50 mM imidazole in 10 mM NaH$_2$PO$_4$, pH 7.5; 150 mM NaCl; 50 mM EDTA; 0.05% surfactant p20, at 30 μl min$^{-1}$, was used to remove non-covalently bound protein, followed by a 180 s injection of 1 M H$_2$N(CH$_2$)$_2$OH, pH 8.5 at 5 μl min$^{-1}$. A sensorgram for CTCB-405 titrated against TbPFK is shown in Supplementary Fig. 6.

**Protein X-ray structural studies**. N-terminally His$_6$-tagged TbPFK was concentrated to 6 mg mL$^{-1}$ and crystallised via hanging drop at 290 K. The well solution consisted of 9.5% PEG, 8000, 0.1 M sodium cacodylate pH 7.4. Crystals formed after 3 weeks. To obtain the complex with CTCB-12, apo-crystals were transferred to a 1 μL drop containing well solution,1 mM CTCB-12 and 1% v/v DMSO for 10 min before being flash cooled directly in liquid nitrogen. Co-crystals of CTCB-360 were obtained from an initial screen using the Molecular Dimensions Morpheus crystallisation screen. Data for complexes of CTCB-12, CTCB-360 and CTCB-405 were collected at the Diamond synchrotron radiation facility on beamlines I03, I04 and I24, respectively. Details of the data collection, structure determination and refinements are given in Supplementary Methods 3.3 and Supplementary Table 2 along with stereo figures (Supplementary Fig. 7) showing the 2|Fo| − |Fc| electron density for the three ligands.

**In vitro parasite killing assays**. The 'in vitro' parasite killing assay used the BSF of the non-human pathogenic subspecies *T. b. brucei*, strain Lister 427, cultured in HMI-9 medium containing 10% foetal bovine serum. A second cell line, *T. b. brucei* GVR35-Luc2 which constitutively expresses firefly luciferase was grown in Modified HMI-9 plus 20% foetal calf serum supplemented with 20% Serum Plus (Sigma) and 0.15 μg mL$^{-1}$ puromycin. Both strains of trypanosomes were cultured in T-25 vented cap flasks at 37 °C and 5% CO$_2$.

The killing time of *T. b. brucei* Lister 427 was assessed using the CellTiter Glo 3D kit (Promega) to measure trypanosome ATP levels as a real-time indicator of viability. Compounds of interest were serially diluted from 16 to 0.5 μM in HMI-9 medium and added into a sterile white, flat bottom 96-well plate (Greiner Inc.). A total of 2500 trypanosomes were added to each well. The plates were incubated at 37 °C and 5% CO$_2$. At the end of each incubation period, CellTiter Glo 3D reagent was added to lyse the trypanosomes and the plates were incubated in the dark for 10 min at RT. Luminescence was measured at a wavelength of 580 nm using a BMG plate reader, and percentage cell viability versus incubation time was determined. Further details of TTK assays for the *T. b. brucei* GVR35-Luc2 strain, the effect of glycerol on the EC$_{50}$ values, and in vitro 'washout' studies to determine the reversibility of the action of the CTCB compounds on trypanosomes are given in Supplementary Methods 4, Supplementary Table 3 and Supplementary Fig. 8.

**In vitro and in vivo pharmacokinetic and efficacy studies**. A HepG2 cytotoxicity assay was used to determine a cellular selectivity index. HepG2 cells were seeded into 96-well plates in EMEM medium. Compounds were assessed for their effect upon cellular viability (Supplementary Methods 5.1). Inhibition of CYP450 was determined using kits from BD Biosciences (Supplementary Methods 5.2). Metabolic stability was determined using mouse and liver microsomes tested at a single concentration of 5 μM. Plasma protein binding was determined using mouse and human plasma at a single concentration of 10 μM (Supplementary Methods 5.4). All PK studies in mice were carried out by Pharmidex (Supplementary Methods 6.1).

All efficacy studies were carried out using CD-1 mice in accordance with the United Kingdom Animals (Scientific Procedures) Act (1986) under Home Office regulations. Studies were performed in SPF facilities at the University of Edinburgh under licence PPL 70/8102 and at the University of Glasgow under PPL 60/4442. The experiments were approved by the local ethics committees of the respective universities. Dosing regimens used in efficacy studies were first tested for tolerance in female CD-1 mice (University of Edinburgh; 6–8 weeks old). Animals were dosed orally by gavage with CTCB compounds following the dosing regimen for each individual experiment ($n = 5$; Supplementary Methods 6.2).

For the mouse model of stage 1 HAT *T. b. brucei* Lister 427 parasites ($1 \times 10^5$) were injected into a single donor female CD-1 mouse (University of Edinburgh; 6–8 weeks old) intraperitoneally in D-PBS containing 10 mM glucose. Parasites for subsequent stage 1 HAT infection were obtained from whole blood via cardiac puncture, under gaseous isoflurane anaesthesia. Parasites ($1 \times 10^3$) harvested from the donor mouse were then injected into the peritoneum of female CD-1 mice (University of Edinburgh; 6–8 weeks old) in D-PBS containing 10 mM glucose. One day after infection, animals were either left untreated ($n = 3$) or dosed orally by gavage ($n = 5$) with CTCB compound in Cremophor EL: EtOH: D-PBS (30:10:60 v/v; Supplementary Methods 6.3).

For the mouse model of stage 1 and stage 2 HAT *T. b. brucei* GVR35-VSL2 parasites ($3 \times 10^4$) were injected into a single donor female CD-1 mouse intraperitoneally. Once infection was established, starting on day 7 for stage 1 studies and day 21 for stage 2 studies, mice were either left untreated ($n = 3$) or dosed by oral gavage with CTCB compound in Cremophor EL: EtOH: D-PBS (30:10:60 v/v; $n = 6$). Progression of *T. b. brucei* GVR35-VSL2 infection was monitored primarily by in vivo bioluminescence imaging of infected mice using an IVIS Spectrum (PerkinElmer). For stage 1 studies, imaging of infected mice was performed before treatment on day 7 post infection and on days following the treatment (days 8, 9 and 14). Imaging on groups of three mice was performed 12 min after intraperitoneal injection of 150 mg kg$^{-1}$ D-luciferin (Promega) in PBS. For stage 2 studies, mice were imaged prior to treatment on day 21, and subsequently on days 22 and 23. On day 23, 24 h after the final dose, mice were sacrificed by cervical dislocation and perfused with PBS containing 15 g L$^{-1}$ glucose to allow ex vivo imaging of whole brains (Supplementary Figs. 12 and 13).

**Reporting summary**. Further information on research design is available in the Nature Research Reporting Summary linked to this article.

## Data availability
Structural X-ray data for protein–ligand complexes are available in the Protein Database with accession numbers 6QU5, 6QU3 and 6QU4. Source data are provided with this paper.

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

## Acknowledgements

This work was supported by the Wellcome Trust (J.C.M.: 104976) and a Wellcome Trust Seeding Drug Discovery award 1017 84/Z/13/Z. We thank the Centre for Translational and Chemical Biology and the Edinburgh Protein Production Facility for use of their facilities. We also thank Professor Keith Matthews for providing *T. b. brucei*, strain Lister *427*.

## Author contributions

Performed in vivo experiments: D.M., L.-H.Y., N.G., E.M. and R.R. Performed biochemical and biophysical experiments: I.W.M., J.K., M.K.W., E.A.B, M.A.W. and P.M.F. Chemical synthesis and design: S.P., A.J.H., A.J.K., A.V. and C.S. Designed experiments, analysed data and wrote paper: E.M., C.S., S.P.W., C.A., J.D., J.C.M., P.A.M.M., S.P. and M.D.W.

## Competing interests

The authors declare no competing interests.
