## [Peer Review File · Nature Communications]

REVIEWER COMMENTS

Reviewer #1 (Remarks to the Author):

Glycosomes are the peroxisome-related organelles exclusively present in Trypanosomatid parasites. The causative agent of African sleeping sickness *Trypanosoma brucei* resides in the host bloodstream and completely relies on glycolysis for the ATP production. Therefore, glycolytic enzymes are considered as potential drug target against these parasites. However, this is challenging since the active sites are highly conserved in these enzymes. Few in vitro studies have described inhibitors of glycolytic enzymes of trypanosomatid parasites before, but they could not reach therapeutic efficacy and/or selectivity.

This manuscript describes novel inhibitors of *T. brucei* phosphofructokinase (TbPFK) which are highly unique in their mechanism of action. These inhibitors selectively bind to parasite-specific allosteric sites in TbPFK and locks the PFK tetramers in an inactive state. These inhibitors rapidly kill parasites and also show very promising therapeutic effect in infected animal models. This structure-based drug design and development study has been performed well with sound methodology and all proper controls. The study shows that even highly conserved metabolic pathways can be targeted by exploiting the differences in allosteric mechanisms between host and pathogens or in proliferative transformations like cancer. Therefore, this manuscript is also of high interest for general audience and suitable for the publication. There are only few minor typographical/technical comments as below.

Minor comments:

Fig. 2 and Fig. 4

Last two sentences in Fig. 4 Legend (regarding pKa) are better suited in the legend of Fig. 2 where pKa and IC50 values are actually mentioned.

Fig. 4

- In the figure, correct PyrroloPyridone to PyrroloPyridine.
- X-and Y-axis should be elaborated in the legend. For e.g. Y-axis reads Tbb_427_gi (EC50, μ M), What does gi indicate?
- Some compounds are labelled in the figure with their CTCB number. The redundant "CTCB-#" in the compound names can be omitted to simplify the figure.

Fig. 5

The legend of Fig. 5a reads "Structure Activity of the pyrrolopyridine scaffold". This should be elaborated for readers if space permits. (For e.g. Three regions in the structure are highlighted in green, blue and yellow, and the respective biological properties or activities are indicated.)

Page 5 line 11 and Table1b (Page 9)

There is discrepancy in the oral bioavailability values for CTCB-508 (112% or 119%)

Page 4 Third paragraph

Cite specific figure sub-panels on Line 7 (Supplementary Figures S2.4.1) and on last Line 11 (Fig. 6a

and Supplementary data S2.4.2)

Fig. 3 (Page 12) and Fig. 6 (Page 15)

Panel headings 'a' and 'b' are missing in the Figure.

Supplementary section 4.5 and Table S4.5 are not referred to in the main text. Since glycerol had little or no influence on EC50, this section can be omitted.

Supplementary Section S1

Authors have tested inhibitor against all three human PFK isoforms in the main text. In Section S1 (last line) only Human PFK-P has been cited. Did authors also purify HsPFK-L and -M or they were purchased commercially? Modify the text accordingly.

S4.1, Line 4

Does "Mod HMI-9" mean "Modified HMI-9"?

S4.2, Line 1

LILIT instead of LILT

Reviewer #2 (Remarks to the Author):

This is a very interesting study describing the development of novel small molecule allosteric inhibitors of trypanosome phosphofructokinase (PFK) using a rational approach that blocks the glycolytic pathway resulting in parasite killing with no inhibition of human PFKs, cross the blood barrier and are effective in reducing parasitemia in infected mice. The implications of the study are that targeting parasite glycolysis and exploring the differences in allosteric mechanisms between trypanosome PFK and human PFKs may be possible to develop specific inhibitors for the treatment of African trypanosomiasis.

The following recommendations are posed to the authors to strengthen this very promising study:

Major:

1. The contention that the allosteric inhibitors, although very promising as a cure for stage 1 sleeping sickness in mice, need to be documented by molecular analysis of the parasite. The results presented in Fig. 7 are very promising using bioluminescence as an initial screening approach. However, to claim that the drugs indeed cure the infected mice, the presence of parasites in mice needs to be evaluated by RT-PCR and mice that survived after drug treatments, need to be immunosuppressed to evaluate if there is a reactivation of parasites by RT-PCR.

Minor:

1. Figure 3: Fig. 3 a and b are not labeled.

2. When the pleomorphic *T. brucei* strain was used, a small percentage of residual parasites were found in the brain, which could be because the late-stage form of the pleomorphic bloodstream-form is less dependent on glycolysis. This needs to be discussed. In addition, the authors should discuss why the pleomorphic strain showed slightly different results than the monomorphic strain.

Reviewer #3 (Remarks to the Author):

This is a nice manuscript. It describes the development of allosteric inhibitors of *Trypanosoma brucei* phosphofructokinase and their assessment as potential therapeutic agents. To my knowledge, this is the first example of in vivo active inhibitors that act specifically against glycolytic enzymes in this family of parasites. The paper outlines the development pathway in a systematic manner, starting with some preliminary hits, moving through lead optimisation involving SAR studies, crystallographic analysis, pharmacology, toxicology, in vitro and in vivo assessment. The work has been competently performed and clearly presented.

There are a number of amendments that I would suggest, and some points that require clarification.

1. I think that the title of the paper overstates the results, given the failure in some cases to achieve parasitological cure. This can be resolved simply by changing the title to ".....can cure stage 1....."
2. In the abstract (lines 57 – 58), the authors state that the compounds can cross the blood brain barrier. For completeness, they should also mention, that although active, the compounds do not achieve sterile cure in this site.
3. In the wash-out experiments (lines 222-227), it seems that the last 1% of the parasites are more difficult to eliminate. Do the authors have any rationale for this?
4. Two different models were used to test in vivo efficacy, CD-1 mice infected with GVR35 and Lister427 parasite strains. Given that the two strains have similar EC50 values (line 280/281), what are the reasons for the differences in cure rate. Is this due to the greater sensitivity of the bioluminescence system (GVR35), or is it that bloodstream parasitemia (in the case of 427) is not an accurate proxy for sterile parasitological cure?
5. Could the authors comment further on the potential (or otherwise) of their compounds to eliminate CNS infections.

Reviewer #4 (Remarks to the Author):

Comments

The authors present a most interesting manuscript describing experimental data, applying and combining complementary bioanalytical and imaging methods, as well as X-ray crystallography as working horse to identify lead compounds, showing evidence for and about efficient acting allosteric functional inhibitors which inhibit the vital Tb phosphofructokinase of different trypanosomia parasites, inhibitors which even can pass the blood brain barrier, able to cure sleeping sickness at an early stage. This results have tremendous impact in the field of drug discovery and drug design investigations to

treat trypanosomia.

The data presented and discussed are based on long term investigations of the authors, previous publications, and successively elaborated experiments. Data and results presented are of high interest for the scientific community working to identify drugs to treat trypanosomia, as well as for the general readership. Figure 7 is showing impressive the success of the investigations and the effectiveness of final selected CTCB compounds.

The manuscript certainly deserves publication

However I have a number of minor comments the authors should consider.

- Even known, however for the general readership a scheme showing the life cycle of the parasite would be informative, option to place in Suppl. Material
- Many abbreviation hamper a little the reading, for example SAR should be written out once, table 1 needs a slightly extended legend for the abbreviations, as other figure and table legends
- In this context CTCB lead compounds identified in terms of previous investigations and following derivatives and evolution and its abbreviations need to be described once more clear. Figure 2 and Figure 4 have a lot of information, however closing the circle is for the readership an issue.
- The authors use the CTCB-12 complex as lead information, however the resolution obtained is rather low and the interpretation "high".
- The authors may describe their crystallization and co-crystallization experiments more clear, S3.3, it is clear the spend lots of efforts to obtain complexes, even applying ligand exchange. Questions they applied TbPFK with His-Tag ? Question, because of the low resolution data for the complex with CTCB-12.
- Technical question, the pdB report, 6QUA, is reporting a complex with CTCB 360, I guess some mismatch happened. Next, for this complex the B-values are very high, beyond 100 Å², the authors may comment on this.
- Figure 3, the zoom box needs to be enlarged, and the authors may consider to place a stereo figure with e-density in the Suppl. Material.
- And minor, the R values reported in the pdb validation report a re slightly different to those reported in Table S3.3 I suggest to have them uniform. And question for the complexes CTCB-360 and CTCB-405 unit cell dimensions a and b are corresponding but interchanged.

Fast acting allosteric inhibitors of phosphofructokinase block trypanosome glycolysis and cure stage 1 sleeping sickness in mice

Submitted to Nature Communications

Reviewer #1 (Remarks to the Author)

Glycosomes are the peroxisome-related organelles exclusively present in Trypanosomatid parasites. The causative agent of African sleeping sickness *Trypanosoma brucei* resides in the host bloodstream and completely relies on glycolysis for the ATP production. Therefore, glycolytic enzymes are considered as potential drug target against these parasites. However, this is challenging since the active sites are highly conserved in these enzymes. Few in vitro studies have described inhibitors of glycolytic enzymes of trypanosomatid parasites before, but they could not reach therapeutic efficacy and/or selectivity.

This manuscript describes novel inhibitors of *T. brucei* phosphofructokinase (TbPFK) which are highly unique in their mechanism of action. These inhibitors selectively bind to parasite-specific allosteric sites in TbPFK and locks the PFK tetramers in an inactive state. These inhibitors rapidly kill parasites and also show very promising therapeutic effect in infected animal models. This structure-based drug design and development study has been performed well with sound methodology and all proper controls. The study shows that even highly conserved metabolic pathways can be targeted by exploiting the differences in allosteric mechanisms between host and pathogens or in proliferative transformations like cancer. Therefore, this manuscript is also of high interest for general audience and suitable for the publication. There are only few minor typographical/technical comments as below.

Minor comments:

Fig. 2 and Fig. 4

Last two sentences in Fig. 4 Legend (regarding pKa) are better suited in the legend of Fig. 2 where pKa and IC50 values are actually mentioned.

Yes; the legends in Figs 2 and 4 have been modified as suggested.

Fig. 4

- In the figure, correct PyrroloPyridone to PyrroloPyridine.

Yes, this has now been corrected in Figure 4.

- X-and Y-axis should be elaborated in the legend. For e.g. Y-axis reads Tbb_427_gi (EC50, μ M), What does gi indicate?

The legend now has clear definitions for both axes also mentioning Supplementary Data S4 where the growth inhibition (gi) assay to measure EC₅₀ values is described.

- Some compounds are labelled in the figure with their CTCB number. The redundant "CTCB-#" in the compound names can be omitted to simplify the figure.

A new Figure 4 has been prepared and simplified as suggested

Fig. 5

The legend of Fig. 5a reads “Structure Activity of the pyrrolopyridine scaffold”. This should be elaborated for readers if space permits. (For e.g. Three regions in the structure are highlighted in green, blue and yellow, and the respective biological properties or activities are indicated.)

This suggested addition to the legend has been incorporated and now reads:

Figure 5a: Structure Activity of the pyrrolopyridine scaffold. Three regions in the structure are highlighted in green, blue and yellow, and the respective biological properties or activities are indicated

Page 5 line 11 and Table 1b (Page 9)

There is discrepancy in the oral bioavailability values for CTCB-508 (112% or 119%)

The values are now consistent (119%)

Page 4 Third paragraph

Cite specific figure sub-panels on Line 7 (Supplementary Figures S2.4.1) and on last Line 11 (Fig. 6a and Supplementary data S2.4.2)

Specific sub panels are now cited as suggested (Figures S2.4.1 and S2.4.2)

Fig. 3 (Page 12) and Fig. 6 (Page 15)

Panel headings ‘a’ and ‘b’ are missing in the Figure.

Panel headings have now been added to both figures

Supplementary section 4.5 and Table S4.5 are not referred to in the main text. Since glycerol had little or no influence on EC50, this section can be omitted.

The role that glycerol plays on parasite metabolism has recently become an interesting topic for parasitologists and given that we have relevant data we have included the following sentence into the main text referring to S4.5

and cell death (15). It has recently been shown that the bloodstream form of *T. brucei* can proliferate on high levels of glycerol, however we have shown that addition of glycerol to the medium has little effect on parasite killing by the CTCB compounds (Supplementary data S4.5).

Authors have tested inhibitor against all three human PFK isoforms in the main text. In Section S1 (last line) only Human PFK-P has been cited. Did authors also purify HsPFK-L and -M or they were purchased commercially? Modify the text accordingly.

We purified all three isoforms of human PFK and tested them to show similar inhibitory activities of the CTCB compounds. The Supplementary text has been changed and a reference for the purification procedures has been added.

The three human isoforms of L, M and P PFK were expressed and purified as described previously (1)

S4.1, Line 4

Does “Mod HMI-9” mean “Modified HMI-9”?

Yes, we have changed text in S4.1 and S4.4 to ‘modified HMI-9’ and included a reference (Myburgh et al, supplementary ref 5) for the full details of this medium.

was grown in Modified HMI-9 plus 20% fetal calf serum (FCS) supplemented with 20% Serum Plus (Sigma) and 0.15 µg/ml puromycin as described previously (5).

S4.2, Line 1

LILIT instead of LILT

Changed to LILIT

Reviewer #2 (Remarks to the Author)

This is a very interesting study describing the development of novel small molecule allosteric inhibitors of trypanosome phosphofructokinase (PFK) using a rational approach that blocks the glycolytic pathway resulting in parasite killing with no inhibition of human PFKs, cross the blood barrier and are effective in reducing parasitemia in infected mice. The implications of the study are

that targeting parasite glycolysis and exploring the differences in allosteric mechanisms between trypanosome PFK and human PFKs may be possible to develop specific inhibitors for the treatment of African trypanosomiasis.

The following recommendations are posed to the authors to strengthen this very promising study:

Major:

1. The contention that the allosteric inhibitors, although very promising as a cure for stage 1 sleeping sickness in mice, need to be documented by molecular analysis of the parasite. The results presented in Fig. 7 are very promising using bioluminescence as an initial screening approach. However, to claim that the drugs indeed cure the infected mice, the presence of parasites in mice needs to be evaluated by RT-PCR and mice that survived after drug treatments, need to be immunosuppressed to evaluate if there is a reactivation of parasites by RT-PCR.

We acknowledge the need for rigour to back this statement; to put this model in context *T. b. brucei* Lister 427 which is used to assess stage 1 efficacy is highly virulent for mice. This monomorphic parasite line will continue to proliferate exponentially in the blood of mice and infection usually requires euthanasia of mice within 4 days of infection in the absence of drug treatment. The standard protocol for this mouse model, as described in the training manual published jointly by the Drugs for Neglected Diseases initiative (DNDi), the Swiss Tropical Institute and the London School of Hygiene and Tropical Medicine (https://assets.publishing.service.gov.uk/media/57a08b6b40f0b652dd00c70/kinetoplastid_drug_screening_manual_181946.pdf) is that a 30 day period of no parasite detection in the blood is considered as cure due to its fast doubling time of ~6hr. Additional confirmation by RT-PCR and immunosuppression is usually only required for assessing cure in mice infected with pleomorphic strains where parasites may persist as stumpy or non-dividing forms in skin, adipose tissue or brain reservoirs. We have modified the text to clarify this mouse model of acute infection for the reader.

Text modified as follows:

These parasites have a doubling time of ~6 hrs (16) and infection with 1×10^5 parasites usually requires euthanasia of mice within 4-5 days of infection in the absence of drug treatment. Drug efficacy is determined by taking regular blood samples from infected mice and checking microscopically for parasitaemia. In this model of acute infection a single surviving parasite should proliferate to reach detectable levels in ~5 days and a 30-day period of no parasite detection is considered as cure.

Minor:

1. Figure 3: Fig. 3 a and b are not labeled.

Labels have been added

2. When the pleomorphic *T. brucei* strain was used, a small percentage of residual parasites were found in the brain, which could be because the late-stage form of the pleomorphic bloodstream-form is less dependent on glycolysis. This needs to be discussed. In addition, the authors should discuss why the pleomorphic strain showed slightly different results than the monomorphic strain.

We have modified the text to explain to the reader why the pleiomorphic strain may be harder to cure. Text modified as follows:

Page 6 Unlike monomorphic Lister 427 *T. b. brucei*, the pleiomorphic GVR35 *T. b. brucei* is controlled by a quorum-sensing mechanism resulting in significant increase in survival time in mice and lower blood parasitaemia that occurs in waves. This extended survival time is also accompanied by parasite dissemination to multiple tissues and organs, including the skin, adipose tissue, lung and brain, with parasite loads increasing over time (17, 19). This more virulent strain usually requires higher or longer doses of trypanocides for cure in vivo compared to the Lister 427 strain (20, 21), possibly caused by reduced drug accessibility in these extravascular tissues.

Reviewer #3 (Remarks to the Author)

This is a nice manuscript. It describes the development of allosteric inhibitors of *Trypanosoma brucei* phosphofruktokinase and their assessment as potential therapeutic agents. To my knowledge, this is the first example of in vivo active inhibitors that act specifically against glycolytic enzymes in this family of parasites. The paper outlines the development pathway in a systematic manner, starting with some preliminary hits, moving through lead optimisation involving SAR studies, crystallographic analysis, pharmacology, toxicology, in vitro and in vivo assessment. The work has been competently

performed and clearly presented.

There are a number of amendments that I would suggest, and some points that require clarification.

1. I think that the title of the paper overstates the results, given the failure in some cases to achieve parasitological cure. This can be resolved simply by changing the title to “.....can cure stage 1.....”

We agree that this is an appropriate change and have modified the title

Fast acting allosteric inhibitors of phosphofructokinase block trypanosome glycolysis and **can cure acute African trypanosomiasis in mice**

2. In the abstract (lines 57 – 58), the authors state that the compounds can cross the blood brain barrier. For completeness, they should also mention, that although active, the compounds do not achieve sterile cure in this site.

We have modified the text to make this point explicit on page 6 para 2:

The brains of saline-perfused mice also showed very clear reduction in parasitaemia (Fig. 7c,d and Supplementary Fig. S6.3.2) confirming compound activity in the CNS **but without achieving sterile parasitological cure.**

3. In the wash-out experiments (lines 222-227), it seems that the last 1% of the parasites are more difficult to eliminate. Do the authors have any rationale for this?

An additional sentence has been added at the end of the section 'The time to kill assay shows very fast parasite killing'.

The existence of a small population of parasites more resistant to the effects of the CTCB compounds might be explained by subpopulations of parasites being present at different stages of the cell cycle. The parasite cultures used in the killing experiments are not synchronised and it is likely that ATP requirement varies depending on cell-cycle phase and a subpopulation with lower ATP requirement may be less susceptible to the CTCB compounds, however this phenomenon needs further investigation.

4. Two different models were used to test in vivo efficacy, CD-1 mice infected with GVR35 and Lister427 parasite strains. Given that the two strains have similar EC50 values (line 280/281), what are the reasons for the differences in cure rate. Is this due to the greater sensitivity of the bioluminescence system (GVR35), or is it that bloodstream parasitemia (in the case of 427) is not an accurate proxy for sterile parasitological cure?

This point is similar to the point made by referee 2 (Minor point 2) and additions to the text have been made to address it in the section 'T. b. brucei Lister 427 infection is cured by one day dosing'

5. Could the authors comment further on the potential (or otherwise) of their compounds to eliminate CNS infections.

With the funding available it was only possible to carry out a limited number of Stage 2 experiments and it might be that increased doses and a modified dosing regimen could indeed clear the parasitaemia. However as alternative highly potent brain penetrant compounds are now moving into clinical trials funding for these next stage experiments has not been possible.

Reviewer #4 (Remarks to the Author):

Comments

The authors present a most interesting manuscript describing experimental data, applying and combining complementary bioanalytical and imaging methods, as well as X-ray crystallography as working horse to identify lead compounds, showing evidence for and about efficient acting allosteric functional inhibitors which inhibit the vital Tb phosphofructokinase of different trypanosomia parasites, inhibitors which even can pass the blood brain barrier, able to cure sleeping sickness at an early stage. This results have tremendous impact in the field of drug discovery and drug design investigations to treat trypanosomia.

The data presented and discussed are based on long term investigations of the authors, previous publications, and successively elaborated experiments. Data and results presented are of high interest for the scientific community working to identify drugs to treat trypanosomiasis, as well as for the general readership. Figure 7 is showing impressive the success of the investigations and the effectiveness of final selected CTCB compounds.

The manuscript certainly deserves publication

However I have a number of minor comments the authors should consider.

- Even known, however for the general readership a scheme showing the life cycle of the parasite would be informative, option to place in Suppl. Material

As suggested by the referee we have included an excellent background reference providing all basic information which is now the first reference in the introduction.

(1) Buscher, G. Cecchi, V. Jamonneau, G. Priotto, Human African trypanosomiasis. *Lancet* **390**, 2397-2409 (2017).

- Many abbreviation hamper a little the reading, for example SAR should be written out once, table 1 needs a slightly extended legend for the abbreviations, as other figure and table legends. In this context CTCB lead compounds identified in terms of previous investigations and following derivatives and evolution and its abbreviations need to be described once more clear. Figure 2 and Figure 4 have a lot of information, however closing the circle is for the readership an issue.

As suggested legends to Table 1 and Figures 2,4 and 5 have been extended and abbreviations explained

Page 2 We used the Structure Activity Relationship (SAR) results

Figure 1 : The legend has been expanded to provide full names for each of the intermediates

Glucose, taken up by the trypanosome from the blood, enters the glycosomes and is converted stepwise via glucose 6-phosphate (glc-6-P), fructose 6-phosphate (fru-6-P), fructose 1,6-bisphosphate (fru-1,6-BP), glyceraldehyde 3-phosphate (G-3-P) and glycerate 1,3-bisphosphate (G-1,3-BP) which is the precursor of 3-phosphoglycerate (3-PGA) that exits in the organelles. The last three steps occur in the cytosol with the conversion of 3-PGA to 2-PGA, the precursor of phosphoenolpyruvate (PEP) which is converted to pyruvate.

Figure 2 The legend has been extended to explain the evolution of the project

CTCB-001, the starting point for this work, was identified from a high throughput screen as a good enzyme inhibitor but with poor trypanocidal activity(10).

Figure 4 has been remade to make it clearer and the legend has been extended (See also comments from referee 1)

The legend to Figure 5 has been extended

Figure 5a: Structure Activity of the pyrrolopyridine scaffold. Three regions in the structure are highlighted in green, blue and yellow, and the respective biological properties or activities are indicated

The authors use the CTCB-12 complex as lead information, however the resolution obtained is rather low and the interpretation "high". The authors may describe their crystallization and co-crystallization experiments more clear, S3.3, it is clear the spend lots of efforts to obtain complexes, even applying ligand exchange. Questions they applied TbPFK with His-Tag ? Question, because of the low resolution data for the complex with CTCB-12.

The referee makes a fair observation regarding the low (3.4 Å) resolution of the CTCB-12 complex however the subsequent series of structures, two of which are presented in this paper, all provide higher resolution structures confirming the binding pose of the CTCB compounds. More experimental detail on the protein purification and crystallisation experiments has been added to S3.3

S3.3 N-terminally His₆-tagged *T. brucei* PFK (TbPFK) was expressed and purified as described previously (1). To obtain apo crystals, purified TbPFK in gel filtration buffer was concentrated to 6 mg/ml and crystallised via hanging drop at 290K. The well solution consisted of 9.5% PEG, 8000, 0.1 M sodium cacodylate pH 7.4. Crystals formed after 3 weeks. To obtain the complex with CTCB-12, apo-crystals were transferred to a 1 µl drop containing well solution, 1mM CTCB-12 and 1% v/v DMSO for 10 minutes before being flash cooled directly in liquid nitrogen.

- Technical question, the pdB report, 6QUA (6qu5), is reporting a complex with CTCB 360, I guess

some mis-match happened. Next, for this complex the B-values are very high, beyond 100 Å², the authors may comment on this.

We thank the author for noticing our typographical error in the pdb title of structure 6qu5: (CTCB-360 should indeed be named CTCB-12). We have written to the pdb to have the 6qu5 structure title changed. The higher than normal B-value for this structure is likely a result of it being such a large structure (twice the size of the other reported structures with two tetramers per asymmetric unit). The crystal was also soaked to introduce the ligand which may also have contributed to the low resolution data and high B-factors.

Figure 3, the zoom box needs to be enlarged, and the authors may consider to place a stereo figure with e-density in the Suppl. Material.

We have expanded Figure 3a with the zoom box and have also prepared a stereo figure showing the electron density for CTCB-12 and also CTCB-360 and CTCB-405. This is referred to in the legend for Figure 3.

Figure S3.3 Walleye stereo drawing of the $2|F_o|-|F_c|$ electron density contoured at 1 sigma round the ligands CTCB-12 (PDB 6QU5, top), CTCB-360 (PDB 6QU3, middle) and CTCB-405 (PDB 6QU4, bottom)

- And minor, the R values reported in the pdb validation report are slightly different to those reported in Table S3.3 I suggest to have them uniform. And question for the complexes CTCB-360 and CTCB-405 unit cell dimensions a and b are corresponding but interchanged.

Values in Table S3.3 have now been corrected and are now consistent. The structures for the complexes with CTCB-360 and CTCB-405 are indeed nearly isomorphous but as the referee correctly points out they do not have the same unit cell naming convention; for 6QU3 $a > b > c$ while for 6QU4 $b > a > c$. However as this does not affect the refinement or structure comparisons, and as both structures are already deposited in the PDB, we feel there is little to be gained by re-indexing.

REVIEWERS' COMMENTS

Reviewer #1 (Remarks to the Author):

All my comments have been appropriately addressed

Reviewer #2 (Remarks to the Author):

The authors have been responsive to the critique. They addressed the major critique raised by this reviewer. Accordingly, the title of the revised manuscript has been changed stating that allosteric inhibitors of phosphofructokinase block trypanosome glycolysis and can cure acute African trypanosomiasis in mice. Thus, this reviewer recommends publication of the revised manuscript.

Fernando Villalta, PhD

Reviewer #3 (Remarks to the Author):

The authors have satisfactorily addressed the issues that I highlighted in my review.

Reviewer #4 (Remarks to the Author):

The authors assigned and considered my questions and comments I raised most well.

They introduced appropriate corrections and additions and modifications.

Technical questions I forwarded are discussed and answered well in the rebuttal.